# Vitamin D and Risk of Obesity-Related Cancers: Results from the SUN (‘Seguimiento Universidad de Navarra’) Project

**DOI:** 10.3390/nu14132561

**Published:** 2022-06-21

**Authors:** Rodrigo Sánchez-Bayona, Maira Bes-Rastrollo, Cesar I. Fernández-Lázaro, Maite Bastyr, Ainhoa Madariaga, Juan J. Pons, Miguel A. Martínez-González, Estefanía Toledo

**Affiliations:** 1Medical Oncology Department, Hospital Universitario 12 de Octubre, 28041 Madrid, Spain; rodrosb@gmail.com (R.S.-B.); ainhoama@hotmail.com (A.M.); 2Department of Preventive Medicine and Public Health, School of Medicine, University of Navarra, 31008 Pamplona, Spain; mbes@unav.es (M.B.-R.); fernandezlazaro@usal.es (C.I.F.-L.); maite.bastyr@gmail.com (M.B.); mamartinez@unav.es (M.A.M.-G.); 3Centro de Investigación Biomédica en Red Área de Fisiología de la Obesidad y la Nutrición (CIBEROBN), INSTITUTO DE SALUD CARLOS III, 28029 Madrid, Spain; 4IdiSNA, Navarra Institute for Health Research, 31008 Pamplona, Spain; jpons@unav.es; 5Department of History, Art History, and Geography, University of Navarra, 31008 Pamplona, Spain

**Keywords:** obesity, cancer, vitamin D, cohort

## Abstract

Obesity is associated with a higher risk of several types of cancer, grouped as obesity-related cancers (ORC). Vitamin D deficiency is more prevalent in obese subjects, and it has been suggested to play a role in the association between obesity and cancer risk. The aim of the study was to analyze the association between vitamin D intake and the subsequent risk of ORC in a prospective Spanish cohort of university graduates. The SUN Project, initiated in 1999, is a prospective dynamic multipurpose cohort. Participants answered a 556-item lifestyle baseline questionnaire that included a validated food-frequency questionnaire. We performed Cox regression models to estimate the hazard ratios (HRs) of ORC according to quartiles of energy-adjusted vitamin D intake (diet and supplements). We included 18,017 participants (mean age = 38 years, SD = 12 years), with a median follow-up of 12 years. Among 206,783 person-years of follow-up, we identified 225 cases of ORC. We found no significant associations between vitamin D intake and ORC risk after adjusting for potential confounders: HR_Q2vsQ1_ = 1.19 (95% CI 0.81–1.75), HR_Q3vsQ1_ = 1.20 (95% CI 0.81–1.78), and HR_Q4vsQ1_ = 1.02 (95% CI 0.69–1.51). Dietary and supplemented vitamin D do not seem to be associated with ORC prevention in the middle-aged Spanish population.

## 1. Introduction

Over recent decades, obesity has become a major public health issue. The prevalence of overweight and obesity has increased in almost all developing and developed countries, reaching nearly 60–70% of the adult population, and being more frequent in women and in urban areas [1,2]. Obesity is commonly defined as a body mass index (BMI) of at least 30 kg/m^2^. The accumulation of excessive fat tissue has been associated with the development of many chronic diseases, most notably hypertension, dyslipidemia, non-alcoholic fatty liver disease, type 2 diabetes, cardiovascular disease, and several types of cancer [3,4,5,6,7]. Obesity constitutes a major determinant for the increasing incidence of cancer, and it could even surpass tobacco as the main preventable cause of cancer [8].

The International Agency for Research on Cancer (IARC) has identified 13 cancers associated with overweight and obesity (grouped under the term of “obesity-related cancers” (ORC)): esophageal adenocarcinoma, postmenopausal breast carcinoma, colon and rectum, uterus, gallbladder, stomach, kidney, liver, ovary, pancreas, thyroid, meningioma, and multiple myeloma [9]. Despite growing evidence, the role of obesity in cancer etiopathogenesis has not been fully elucidated. The main mechanisms that seem to be implicated in the association between obesity and cancer are hyperinsulinemia, subclinical chronic low-grade inflammation, alterations in adipocytokine pathophysiology, and hormonal imbalance [10,11]. 

Vitamin D deficiency is a global health problem. For adults aged 19–80 years, the recommended dietary intake of vitamin D is between 10 and 20 µg/day [12]. Approximately 60% of adults worldwide are vitamin D deficient. Although several factors may explain the high prevalence of low vitamin D levels, inadequate sun exposure (i.e., indoor environment, excess of sun avoidance, air pollution) and low vitamin D intake (i.e., dietary lifestyles, lactose intolerance, and even socio-economic status) are the most common causes [13]. Dietary sources of vitamin D include oily fish (such as salmon and tuna fish), red meat, liver, egg yolks, dairy products, and cereals.

Vitamin D is one of many factors suggested to play a role in the obesity-cancer pathway. Vitamin D can be considered as a mediator, an effect modifier, or a confounder in the association between obesity and higher risk of these cancers. Previous studies have found an association between obesity and vitamin D deficiency, although it remains unclear whether vitamin D deficiency leads to an altered metabolism, or the altered metabolic state of obesity leads to vitamin D deficiency. A higher prevalence of vitamin D deficiency in the obese population may be explained through different mechanisms. One mechanism may combine lower dietary intake with lower sunlight exposure or impaired cutaneous vitamin D synthesis. Another mechanism may be influenced by differences in protein binding and metabolic clearance in obese people that could cause lower levels of circulating vitamin D [14,15]. Some observational studies have also suggested that deficient vitamin D levels contribute to a higher risk of malignant neoplasia, such as breast and colorectal cancer [16,17]. However, the question of whether vitamin D influences the association of obesity with cancer (i.e., whether it acts as an effect modifier) has not been prospectively addressed. The available evidence is currently insufficient to be able to support the supplementation of vitamin D as a treatment strategy to mitigate the negative effects its deficiency may have on cancer incidence and survival.

In this study, we aimed to assess the impact of vitamin D intake on the subsequent risk of obesity-related cancers in a prospective Spanish cohort of university graduates. 

## 2. Materials and Methods

### 2.1. Study Population

The ‘Seguimiento Universidad de Navarra’ (SUN) Project is an ongoing multipurpose cohort study composed of university graduates in Spain [18]. The cohort recruitment began in 1999 and is ongoing. In the entire cohort, the median age at recruitment was 34.7 years (interquartile range: 26–42 years) and 61% of the participants are female. When participants were recruited, they completed a baseline 556-item questionnaire, collecting information about lifestyle, sociodemographic, anthropometric, and medical variables. After completing the baseline questionnaire, participants were contacted biennially through follow-up questionnaires to collect information on lifestyle changes and incident medical conditions. 

Through December 2019, a total of 22,894 participants completed the baseline questionnaire. For this analysis, we used the following exclusion criteria: 341 participants who answered the baseline questionnaire after 1 March 2017 were excluded to assure a minimum 2-year follow-up period; we further excluded 1889 participants lost in follow-up (overall retention 92%); we also excluded 540 participants with a previous cancer diagnosis at the time of enrollment. Lastly, we excluded 1891 participants with energy intake outside of predefined limits (a daily energy intake below 500 kcal/d or above 3500 kcal/d for women and below 800 kcal/d or above 4000 kcal/d for men) [19], and 216 participants with extreme intake of vitamin D (+/– 3 standard deviations). Finally, a total of 18,017 participants were included (Figure 1). The Institutional Review Board of the University of Navarra approved this study.

### 2.2. Assessment of Vitamin D Intake

At baseline, participants completed a validated 136-item semi-quantitative food-frequency questionnaire (FFQ). The reproducibility and validity of this FFQ has been previously published by our group [20]. The questionnaire gathers information from a wide variety of food groups, such as high-fat dairy products, eggs, meat, fish, seafood, vegetables, fruits, cereals, legumes, processed pastries, or fast food, among others. For each item, a commonly used portion size is defined. Participants were asked to provide the information in terms of long-term dietary exposures. Information was also gathered on the regular use of supplements or multivitamins, including brand, dosage, and frequency. For each subject, energy and nutrient intakes were calculated using food composition tables [21,22]. The total estimated vitamin D intake combined both diet and supplements. Total and dietary vitamin D intake were adjusted for total energy intake with the residual method [19]. As previously demonstrated for our cohort, the FFQ provides a good reproducible assessment of the usual diet and a reasonable validity in relation to vitamin D (intraclass correlation coefficient (ICC) = 0.69, using repeated 3-day dietary records as reference) [23]. For these analyses, participants were categorized into quartiles according to dietary or total (dietary plus supplemented) vitamin D intake. 

### 2.3. Ascertainment of Obesity-Related Cancer Cases

In our study, we considered the outcome of interest to be all the incident cases of any of the following cancer diagnoses: esophageal adenocarcinoma, postmenopausal breast carcinoma, colon and rectum, uterus, gallbladder, stomach, kidney, liver, cholangiocarcinoma, ovary, pancreas, thyroid, meningioma, and multiple myeloma. Initially, cancer cases were self-reported. Participants who reported a diagnosis of any tumor were then asked to provide a copy of their medical records. Subsequently, an independent expert oncologist, who was blinded to the exposure, confirmed the cases by reviewing these records. If any participant did not submit a medical record, they were asked to consent to be contacted via telephone by an expert to confirm malignancy. Deaths due to any cancer identified by reviewing the National Death Index (NDI) were also included as confirmed cases.

### 2.4. Covariate Assessment

The baseline questionnaire collected information on sociodemographic, lifestyle, and medical variables. The self-reported accuracy of height and weight to estimate the BMI has been previously validated in this cohort [24]. Physical activity was also assessed through a validated questionnaire [25]. The Mediterranean Diet Score proposed by Trichopoulou et al. [26] was used to evaluate the adherence to the Mediterranean dietary pattern, excluding alcohol intake. We considered alcohol consumption as a separate covariate given the growing evidence on its association with several ORC [27,28]. The questionnaire also gathered information on the participants’ average time of sunlight exposure. Participants were inquired about the hours/day of sunlight exposure during the week and for a typical day during the weekend in winter and during the summer. In order to estimate a proxy of solar irradiation intensity in the location of residence, we consulted the Global Horizontal Irradiance (GHI, kWh/m^2^/day) in a given postal area between 1994 and 2015 and linked this to the participants’ postal codes. Information about the GHI can be obtained from the Global Solar Atlas satellite-derived dataset [29]. Missing values were imputed (simple imputation) using the Stata built-in command *impute*, based on multivariable linear regression models for continuous variables and multivariable logistic or multinomial regression models for categorical variables. Imputations represented <5% of missing covariates, except for tobacco consumption (pack-years) with 10% of missing values.

### 2.5. Statistical Analysis

Baseline characteristics of participants were described according to quartiles of total vitamin D intake. Quantitative variables were summarized with means and standard deviations, and qualitative variables with proportions. We verified the normality of distribution with the Shapiro–Wilk test. We used the ANOVA test to compare quantitative traits across quartiles of total vitamin D intake, and we reran these analyses using Kruskal–Wallis tests for those variables with a non-parametric distribution. We also used the Chi-squared test to compare qualitative traits across quartiles of total vitamin D intake. 

To examine the association of vitamin D intake (diet or diet and supplements) and the risk of ORC, we fitted Cox regression models, with the lowest quartile as the reference category. The models included age as an underlying time variable and were additionally stratified by recruitment period and age (decades) at recruitment. The time at entry was defined as the date of completion of the baseline questionnaire. The outcome was defined as the date of ORC diagnosis. For exit time, we considered the age of cancer diagnosis for cases and the date of death due to a non-ORC-related cause, or lost to follow-up, for non-cases. We adjusted a first multivariable model, including the following potential confounders: sex, height (cm), family history of breast or colorectal cancer (yes/no), smoking habit (never, current, or former smoker), lifetime tobacco consumption (pack-years), years of university studies, physical activity (METs-h/week), alcohol consumption (g/day), total energy intake (kcal/day), BMI (kg/m^2^), consumption of sugar-sweetened beverages (servings/day), coffee consumption (servings/day), TV-watching (h/day), sunlight exposure (h/year), and intensity of solar irradiation in the residential area (kWh/m^2^/day). In a second multivariable model, we additionally adjusted for adherence to Mediterranean Diet Score (0–8 points). We selected potential confounders based on existing evidence and previous results of our cohort studies on cancer [30,31,32,33,34,35]. We estimated the association with ORC risk for both total vitamin D intake (dietary and supplemented) and dietary vitamin D as main exposures. Additionally, to analyze the potential modification of the effect of vitamin D intake by BMI categories (normal weight and overweight/obesity), we performed the multivariable Cox regression model according to BMI strata. We studied multiplicative interaction between vitamin D intake quartiles and BMI using the likelihood-ratio test to assess for the statistical significance of a product term. As sensitivity analysis, we repeated our analyses excluding those tumors which accounted for at least 10% of overall cases.

Analyses were performed using STATA/SE version 15.0 (StataCorp). A two-sided *p* value < 0.05 was deemed as statistically significant. 

## 3. Results

For the analysis, we included 18,017 participants, with a median follow-up of 12.2 years. Table 1 describes baseline characteristics of participants according to quartiles of total (dietary and supplemented) energy-adjusted vitamin D intake. The median BMI was 23.1 kg/m^2^ (interquartile range: 20.9–25.6 kg/m^2^). In our cohort, only 25% of participants met the recommended dietary intake of vitamin D for adults 19–80 years. Participants in the highest quartile of vitamin D intake tended to be more physically active, to have a slightly higher adherence to the Mediterranean diet, and to have a higher sunlight exposure (in h/year). Other important characteristics such as sex, BMI, total energy intake, family history of breast or colorectal cancer, and sunlight exposure were very similar across groups. 

During a total follow-up of 206,783 person-years, we identified 225 ORC cases (59 postmenopausal breast, 49 colon and 21 rectum, 11 uterus, 6 ovarian, 22 pancreatic, 5 esophageal, 11 stomach, 2 gallbladder, 7 cholangiocarcinoma, 4 hepatocellular carcinoma, 4 multiple myeloma, 1 meningioma, 9 renal cell carcinomas, and 21 thyroid carcinomas) (7 participants developed two ORC).

When we compared quartiles of total vitamin D intake, we found no significant associations for ORC risk after adjusting for potential confounders (Table 2). Compared to the lowest quartile, the HR_Q2vsQ1_ was 1.19 (95% CI 0.81–1.75), the HR_Q3vsQ1_ was 1.20 (95% CI 0.81–1.78), and the HR_Q4vsQ1_ was 1.02 (95% CI 0.69–1.51). 

When comparing the risk of ORC across quartiles of dietary vitamin D intake (using the lowest quartile of vitamin D intake as the reference category), we found no significant association (Table 3): HR_Q2vsQ1_ = 1.14 (95% CI 0.77–1.69), the HR_Q3vsQ1_ = 1.25 (95% CI 0.84–1.87), and the HR_Q4vsQ1_ = 1.08 (95% CI 0.73–1.61).

When we excluded tumors representing at least 10% of cases (colorectal, breast, pancreatic, and thyroid carcinomas) results did not change substantially (Figure 2).

The effect of total vitamin D intake in the ORC risk did not vary across BMI strata (Figure 3). We found no interaction between quartiles of vitamin D intake and BMI categories (p for interaction = 1.00) in the subsequent risk of ORC. 

## 4. Discussion

In this prospective Spanish cohort, the risk of ORC did not significantly change across quartiles of vitamin D intake, regardless of considering total or dietary intake.

Evidence supporting a link between obesity and cancer, and additionally between low vitamin D and obesity, is consistent across studies. A pooled analysis of 12 observational studies that evaluated the association between vitamin D status and obesity showed an overall relative risk of 1.52 (95% CI 1.33–1.73) for low vitamin D status and obesity [36]. This inverse association may be attributed to an increase in metabolic clearance in the excess adipose tissue characteristic in obese states [37]. Another explanation could be that obese individuals are less likely to engage in outdoor physical activity than non-obese individuals, decreasing sun exposure [38]. The benefit of leisure-time physical activity in reducing the risk of breast cancer has been previously explored in the SUN cohort [35]. In terms of physical activity, the SUN cohort participants have a median of 16.1 METs-h/week. This represents approximately an equivalent of 1 h/day of daily walking. In addition, the mean BMI of the participants included in our analysis was 23.5 kg/m^2^ (SD = 3.5 kg/m^2^). Some baseline characteristics of the population included in our study (moderate/high physical activity, middle-aged participants, and mean BMI in the normal weight range) could outweigh the potential role of vitamin D intake in reducing the risk of cancer. 

In the present study, 131 out of 225 cases (58%) of ORC identified in the SUN cohort were composed of postmenopausal breast cancer and colorectal cancer. Among all types of ORC, colorectal cancer has shown more consistent results for an inverse association between vitamin D levels and cancer risk. Observational studies performed in different populations worldwide have found a relative decrease in colorectal cancer risk varying from 4 to 50% for the comparison of extreme categories [39,40,41]. As for breast cancer risk, while some observational studies have reported an inverse association between vitamin D intake and breast cancer, others have reported null associations, leading to inconclusive evidence concerning vitamin D intake [42,43]. Previously in our cohort, we specifically analyzed the association between dietary calcium, vitamin D, and breast cancer risk [44]. We found a non-linear association between total calcium intake and breast cancer, with risk reductions associated with higher intake up to approximately 1400 mg/day. No evidence for any association between vitamin D intake and breast cancer was found (HR_T3vsT1_ = 0.87, 95% CI 0.54–1.41). 

The potential role of vitamin D intake in cancer prevention was hypothesized several decades ago [45]. Results from observational studies led to the idea that vitamin D deficiency may increase cancer risk. However, vitamin D supplementation in interventional studies has not shown cancer preventive qualities. In the VITAL trial, conducted in the United States, more than 25,000 participants received a daily vitamin D3 supplementation of 2000 IU (50 mcg) for 5 years, and no significant reduction in overall cancer risk was found [46]. Other randomized controlled trials have also reported no effect of supplementing vitamin D3 in cancer risk [47,48,49]. A differential effect of vitamin D intake across populations has been suggested through studies analyzing single nucleotide polymorphisms associated with 25(OH)D3 levels [50]. Studies performed in the Finnish population have suggested that the possible beneficial effects of vitamin D3 on cancer may not be consistent in the general population. Hence, there could exist ‘low vitamin D responders’—individuals who need to increase their dose of daily vitamin D supplementation to reach full clinical benefit—in contrast with ‘high vitamin D responders’—those who can have a vitamin D deficiency and might be less affected by diseases, such as malignant neoplasms, against which vitamin D may have a preventive role [51,52]. 

Dietary intake and supplementation of vitamin D may prevent pro-inflammatory processes, such as metabolic syndrome and carcinomas. Despite promising results from previous studies, vitamin D cannot be considered an anti-cancer agent yet, as its potential anti-tumoral activity has not been fully confirmed. Similar to many other nutritional co-factors, such as vitamins or polyphenols, vitamin D can exert a pleiotropic role within cellular machinery with the ability to promote cell response to stress [53]. The active form of vitamin D (1α,25(OH)_2_vitD or calcitriol) can be considered a dietary-derived immune cytokine. It may interact with immune system, as lymphocytes express the vitamin D receptor (VDR) [54]. However, it should be noted that most of the immunological activity exerted by calcitriol depends on cellular VDR expression. The VDR polymorphisms must be considered when assessing the efficacy of vitamin D in counteracting cancer risk in a given population [55]. In our analyses, we included many potential confounders because any research on the nutritional value expected from vitamin D supplementation has to consider many factors, including lifestyle, different dietary habits from diverse regions, and metabolic state [56]. 

Some limitations must be noted. First, our cohort participants are relatively young and physically active and show a mean BMI mostly in the normal weight range. These characteristics may partially explain the low incidence of cancer. Consequently, this may limit the statistical power, especially when analyzing the association with each specific type of neoplasia separately. Additionally, to be noted is the fact that only 25% of participants included in the analyses met the recommended dietary intake of vitamin D (10–20 µg/day). This may have limited the capacity to observe the influence of this nutrient (being consumed in adequate amounts). Second, it is important to note that, under the concept of ORC, we include tumors with different pathobiology, carcinogenesis, and cellular pathways. Hence, the biological heterogeneity may also limit the capacity to draw solid conclusions. Third, as the exposure is self-reported and participants may misreport their nutritional pattern in the questionnaire, this potential bias could result in the observed association being underestimated. Fourth, the exposure period considered may not be the most etiologically relevant for participants, as dietary patterns may differ from the early adulthood. Fifth, since the outcome assessment was self-reported, this may have resulted in an underreporting of incident cases and thus a lower sensitivity. Nevertheless, cancer cases were blindly confirmed—with high specificity—by an oncologist. Sixth, since the socio-economic status of the participants was not available, years of university studies were included in the multivariable analysis. As our study sample exclusively involved university graduates, it is homogeneous in this aspect, which may reduce the potential confounding effect of educational and socioeconomic status.

To the best of our knowledge, this is the first study to assess the relationship between vitamin D intake and ORC risk in a middle-aged Spanish population. The prospective nature of the SUN Project ensures the temporal sequence between exposure and outcome, including a large sample size with a long follow-up and a good retention rate. Moreover, the adjustment for a wide number of potential confounders and the sensitivity analyses assure the robustness of our findings. Self- reported cancer cases were confirmed via medical reports to ensure that the final diagnosis was an invasive carcinoma and not a benign lesion.

## 5. Conclusions

In summary, our study did not find any association between vitamin D intake and risk of ORC. Dietary and supplemental vitamin D do not seem to be associated with ORC prevention in the middle-aged Spanish population. Current guidelines for reducing cancer risk should focus on the detrimental effect of obesity or smoking, and promote other healthy habits such as regular exercise, weight loss, and adherence to the Mediterranean diet, as they have demonstrated more consistent evidence as preventive measures.

## Figures and Tables

**Figure 1 nutrients-14-02561-f001:**
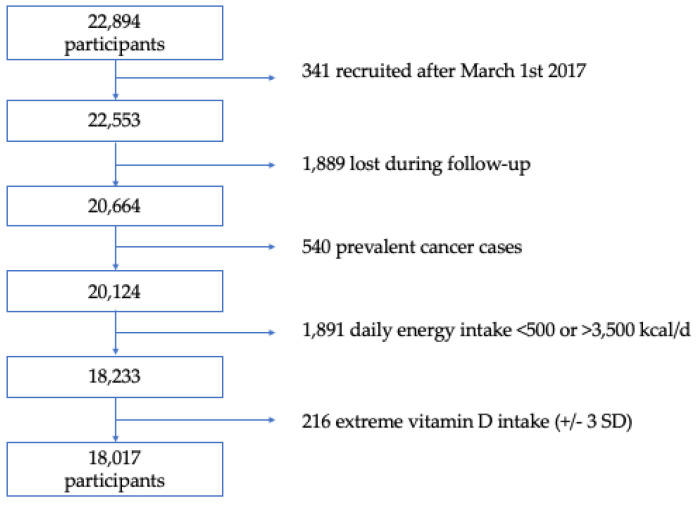
Flowchart of participants in the SUN Project, 1999–2019. Kcal/d: kilocalorie per day. SD: standard deviation.

**Figure 2 nutrients-14-02561-f002:**
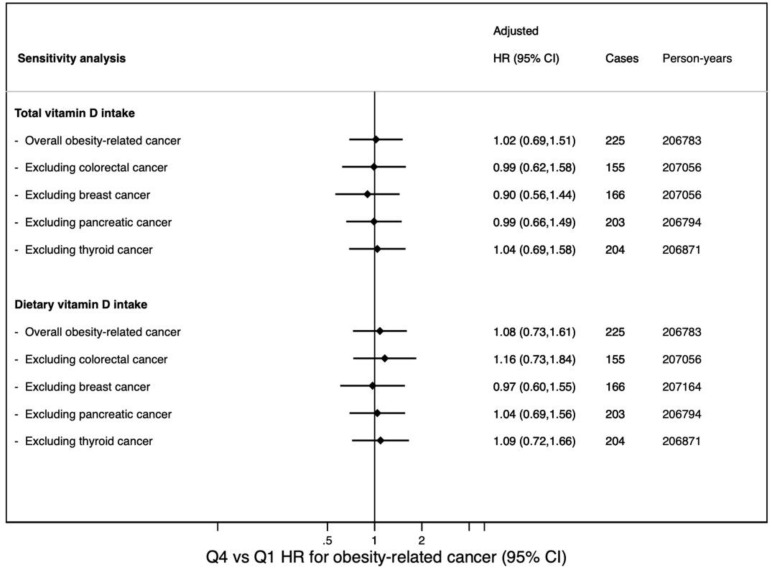
Hazard ratio (95% CI) for the comparison across extreme quartiles of overall obesity-related cancer and excluding tumors which accounted for at least 10% of cases.

**Figure 3 nutrients-14-02561-f003:**
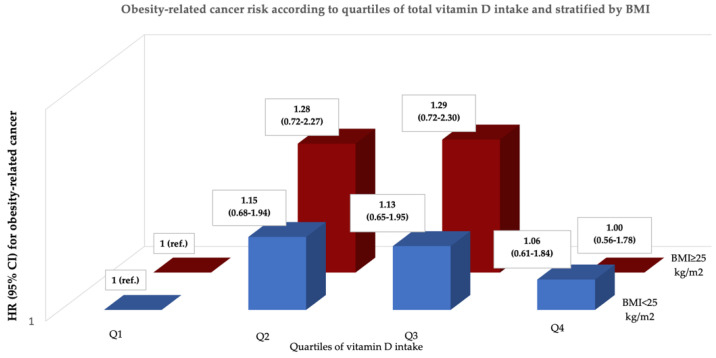
Hazard ratio (95% CI) of obesity-related cancer according to quartiles of total vitamin D intake and BMI strata (normal weight and overweight/obesity). Adjusted for sex, height (cm), family history of breast or colorectal cancer (yes/no), smoking habit (never, current, or former smoker), lifetime tobacco consumption (pack-years), years of university studies, physical activity (MET-h/week), alcohol consumption (g/day), total energy intake (kcal/day), consumption of sugar-sweetened beverages (servings/day), coffee consumption (servings/day), TV-watching (h/day), sunlight exposure (h/year), intensity of solar irradiation (kWh/m^2^/day), and adherence to Mediterranean Diet Score (0–8 points).

**Table 1 nutrients-14-02561-t001:** Baseline characteristics of participants in the SUN Project, according to energy-adjusted quartiles of vitamin D (diet and supplemented).

Variable	Q1	Q2	Q3	Q4	*p* Value ^+^
n	4505	4504	4504	4504	
Total vitamin D intake (µg/day) *	2.7 (2.0–3.2)	4.4 (4.0–4.7)	5.8 (5.4–6.5)	11.7 (11.0–12.5)	<0.001
Age (years)	35 (27–45)	36 (27–46)	35 (27–46)	37 (28–49)	<0.001
Sex (% women)	62.1	59.3	58.4	59.6	0.003
Body-mass index (kg/m^2^)	22.8 (20.7–25.4)	23.1 (20.9–25.6)	23.1 (20.9–25.5)	23.4 (21.0–25.9)	<0.001
Height (cm)	168 (162–174)	168 (162–175)	168 (162–175)	168 (162–174)	0.174
Physical activity (METs-h/week)	14.2 (4.2–27.5)	14.8 (5.1–29.1)	15.9 (5.7–29.8)	18.9 (7.4–34.4)	<0.001
Total energy intake (kcal/day)	2517 (2048–2997)	2362 (1976–2715)	2041 (1682–2503)	2358 (1975–2767)	<0.001
Alcohol intake (g/day)	2.6 (0.6–8.8)	3.2 (0.9–8.8)	3.1 (0.9–8.4)	3.2 (0.9–9.0)	<0.001
Sugar-sweetened beverages (servings/day)	0.1 (0.0–0.4)	0.1 (0.0–0.1)	0.1 (0.0–0.1)	0.1 (0.0–0.1)	<0.001
Coffee (servings/day)	1.0 (0.4–2.5)	1.0 (0.4–2.5)	1.0 (0.4–2.5)	1.0 (0.4–2.5)	<0.001
Adherence to Mediterranean Diet Score	4 (3–5)	4 (3–5)	4 (3–5)	5 (3–6)	<0.001
Time of university education (years)	5 (4–5)	5 (4–5)	5 (4–5)	5 (4–5)	0.047
Smoking habit (%)					<0.001
Never	47.6	48.7	50.6	47.7	
Current	23.8	22.6	21.5	20.2	
Former	28.6	28.7	27.9	32.1	
Tobacco consumption (pack-years)	0.5 (0.0–10.0)	0.5 (0.0–10.0)	0.0 (0.0–9.0)	0.5 (0.0–11.0)	<0.001
TV watching (h/day)	1.5 (0.8–2.0)	1.5 (0.8–2.0)	1.4 (0.7–2.0)	1.4 (0.8–2.0)	0.101
Solar irradiation (kWh/m^2^/day)	4.0 (3.9–4.5)	4.0 (3.7–4.6)	4.1 (3.8–4.6)	4.2 (3.7–4.7)	0.192
Sunlight exposure (h/year)	1984 (1866–2816)	1984 (1866–2822)	1984 (1866–2887)	2279 (1866–2887)	<0.001
Family history (%)					
Breast cancer	27.0	25.6	27.4	27.1	0.21
Colorectal cancer	14.6	14.6	15.1	16.3	0.09

* Values represent medians (interquartile range), unless otherwise stated. ^+^ Chi-squared test for comparisons of proportions and ANOVA test or Kruskal–Wallis test for quantitative traits.

**Table 2 nutrients-14-02561-t002:** Hazard ratio (95% CI) of obesity-related cancer according to energy-adjusted quartiles of total vitamin D intake (dietary and supplemented).

Obesity-Related Cancer Cases	Q1	Q2	Q3	Q4
Cases/person-years	50/52657	57/52666	58/51328	60/50131
Age adjusted	1 (Ref.)	1.14 (0.78–1.67)	1.14 (0.78–1.67)	1.01 (0.69–1.48)
Multivar. adjusted *	1 (Ref.)	1.19 (0.81–1.76)	1.21 (0.82–1.79)	1.06 (0.72–1.55)
Multivar. adjusted ^†^	1 (Ref.)	1.19 (0.81–1.75)	1.20 (0.81–1.78)	1.02 (0.69–1.51)

* Adjusted for sex, height (cm), family history of breast or colorectal cancer (yes/no), smoking habit (never, current, or former smoker), lifetime tobacco consumption (pack-years), years of university studies, physical activity (MET-h/week), alcohol consumption (g/day), total energy intake (kcal/day), BMI (kg/m^2^), consumption of sugar-sweetened beverages (servings/day), coffee consumption (servings/day), TV-watching (h/day), sunlight exposure (h/year), and intensity of solar irradiation (kWh/m^2^/day). ^†^ Additionally adjusted for adherence to Mediterranean Diet Score (0–8 points).

**Table 3 nutrients-14-02561-t003:** Hazard ratio (95% CI) of obesity-related cancer according to energy-adjusted quartiles of dietary vitamin D intake.

Obesity-Related Cancer Cases	Q1	Q2	Q3	Q4
Cases/person-years	48/52523	55/52640	59/51412	63/50205
Age adjusted	1 (Ref.)	1.12 (0.76–1.66)	1.19 (0.81–1.75)	1.07 (0.73–1.56)
Multivar. adjusted *	1 (Ref.)	1.15 (0.78–1.71)	1.27 (0.85–1.89)	1.12 (0.76–1.65)
Multivar. adjusted ^†^	1 (Ref.)	1.14 (0.77–1.69)	1.25 (0.84–1.87)	1.08 (0.73–1.61)

* Adjusted for sex, height (cm), family history of breast or colorectal cancer (yes/no), smoking habit (never, current, or former smoker), lifetime tobacco consumption (pack-years), years of university studies, physical activity (MET-h/week), alcohol consumption (g/day), total energy intake (kcal/day), BMI (kg/m^2^), consumption of sugar-sweetened beverages (servings/day), coffee consumption (servings/day), TV-watching (h/day), sunlight exposure (h/year), and intensity of solar irradiation (kWh/m^2^/day). ^†^ Additionally adjusted for adherence to Mediterranean Diet Score (0–8 points).

## Data Availability

The data that support the findings of this study are available from the SUN Project at sun@unav.es, upon reasonable request.

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
