# Peer review of "Vitamin D and Risk of Obesity-Related Cancers: Results from the SUN (‘Seguimiento Universidad de Navarra’) Project"

_nutrients, 2022, doi:10.3390/nu14132561_

Round 1
Reviewer 1 Report
The manuscript entitled „Vitamin D and risk of obesity-related cancers: results from the SUN (‘Seguimiento Universidad de Navarra’) Project” presents interesting issue, but some issues should be corrected.
Major:
Based on the referred publications it seems that SUN project was conducted in Spanish population only, while in the presented manuscript, it is introduced as Mediterranean population. Authors should clearly define that the study was conducted in a national population, no in the international one.
Abstract:
Instead of what was done („We analyzed…”) Authors should formulate what was the aim of the study (e.g. „The aim of the study was…”).
Authors should avoid too general conclusions – they did not study international population, so it should be reflected.
Introduction:
In the whole manuscript Authors use term “obesity-related cancers” (ORC), which in the Introduction Section is not properly defined. While introducing this term, Authors refer the publication by Hopkins et al. (https://www.ncbi.nlm.nih.gov/pmc/articles/PMC5562429/) in which this term is not used. Authors should either present clear definition, or not use such term if they are not able to define it properly.
Authors should deepen the issue of vitamin D deficiency – how frequent it is, why is it so frequent, what are the major sources of vitamin D, etc.
Materials and Methods:
The population which was studied within the SUN Project should be clearly presented and described
Authors should describe the FFQ which they used – which food products were included, what was the reproducibility and validity for vitamin D, etc.
Authors should clearly describe how did they assess sunlight exposure – did they use any validated questionnaire, or just asked a single question. If it was a single question, it should be presented here.
It seems that Authors did not verify normality of distribution.
Authors should verify normality of distribution and only for parametric data they should present mean and SD, while for non-parametric they should present median, min and max values.
Authors should use statistical tests based on the distributions observed.
Results:
Authors should present clearly the vitamin D intake in the studied group, including the share of respondents meeting the recommended dietary intake.
Table 1 – the values should be compared between quartiles
It seems that Authors did not verify normality of distribution.
Authors should verify normality of distribution and only for parametric data they should present mean and SD, while for non-parametric they should present median, min and max values.
Authors should use statistical tests based on the distributions observed.
Discussion:
Authors should reflect here the problem of meeting the recommended dietary intake of vitamin D. The problem in their group may result from the fact that the majority of respondents may have in fact inadequate intake of vitamin D. If so, they were not able to observe the influence of this nutrient (being consumed in adequate amount).
Conclusions:
Authors should avoid too general conclusions – they did not study international population, so it should be reflected.
Reviewer 2 Report
Dear Authors,
Thank you for your manuscript. It is well-designed and well-written. The study is professionally designed according to epidemiological standards. Please consider my comments and concerns below.
Abstract. Please revise the sentence in lines 25-26: "We found no significant associations for ORC risk after adjusting for potential confounders". Associations between vitamin D intake and ORC risk?
Introduction. The authors state: "Previous works have found an association between obesity and vitamin D deficiency" (lines 52-53). Please explain more what reason(s) could lead to this association?
And for me, it is strange and unusual testing the association between vitamin D intake and ORC risk in young, active and non-obese study subjects. Could the authors provide % of under-, normal-weight, overweight and obese subjects in the cohort at the baseline? Also, could the authors provide a comparison of vitamin D intake in BMI groups?
Also, for applying the Cox regression, the exact date of the outcome in the cohort is required. Please specify this in section 2.3.
When reading, I noticed the same study limitations as the Authors have stated, so I will not be sticking to the details anymore. Also, as a strength, I see the decision to verify self-reported cancer diagnoses by the oncologist.
All the best.
Round 2
Reviewer 1 Report
The manuscript entitled „Vitamin D and risk of obesity-related cancers: results from the SUN (‘Seguimiento Universidad de Navarra’) Project” presents interesting issue, but some issues should be corrected.
Materials and Methods:
Authors should verify normality of distribution and only for parametric data they should present mean and SD, while for non-parametric they should present median, min and max values.
Authors should use statistical tests based on the distributions observed.
Results:
Authors should present clearly the vitamin D intake in the studied group, including the share of respondents meeting the recommended dietary intake.
